# Neurophysiological Oscillatory Mechanisms Underlying the Effect of Mirror Visual Feedback-Induced Illusion of Hand Movements on Nociception and Cortical Activation

**DOI:** 10.3390/brainsci14070696

**Published:** 2024-07-12

**Authors:** Marco Rizzo, Laura Petrini, Claudio Del Percio, Lars Arendt-Nielsen, Claudio Babiloni

**Affiliations:** 1Center for Neuroplasticity and Pain (CNAP), SMI®, Department of Health Science and Technology, Aalborg University, 9220 Aalborg, Denmark; marco.rizzo@mssm.edu (M.R.); lap@hst.aau.dk (L.P.); lan@hst.aau.dk (L.A.-N.); 2Department of Physiology and Pharmacology “V. Erspamer”, Sapienza University of Rome, 00185 Rome, Italy; claudio.delpercio@uniroma1.it; 3Department of Medical Gastroenterology, Mech-Sense, Aalborg University Hospital, 9220 Aalborg, Denmark; 4Hospital San Raffaele Cassino, 03043 Cassino, Italy

**Keywords:** high-density electroencephalography (HD-EEG), alpha event-related de/synchronization (ERD/ERS), source analysis, Mirror Visual Feedback (MVF), sensory–motor interaction, pain

## Abstract

Mirror Visual Feedback (MVF)-induced illusion of hand movements produces beneficial effects in patients with chronic pain. However, neurophysiological mechanisms underlying these effects are poorly known. In this preliminary study, we test the novel hypothesis that such an MVF-induced movement illusion may exert its effects by changing the activity in midline cortical areas associated with pain processing. Electrical stimuli with individually fixed intensity were applied to the left hand of healthy adults to produce painful and non-painful sensations during unilateral right-hand movements with such an MVF illusion and right and bilateral hand movements without MVF. During these events, electroencephalographic (EEG) activity was recorded from 64 scalp electrodes. Event-related desynchronization (ERD) of EEG alpha rhythms (8–12 Hz) indexed the neurophysiological oscillatory mechanisms inducing cortical activation. Compared to the painful sensations, the non-painful sensations were specifically characterized by (1) lower alpha ERD estimated in the cortical midline, angular gyrus, and lateral parietal regions during the experimental condition with MVF and (2) higher alpha ERD estimated in the lateral prefrontal and parietal regions during the control conditions without MVF. These preliminary results suggest that the MVF-induced movement illusion may affect nociception and neurophysiological oscillatory mechanisms, reducing the activation in cortical limbic and default mode regions.

## 1. Introduction

The phenomenon whereby the mirrored movement of one limb is perceived as the simultaneous movement of the opposite limb is known as Mirror Visual Feedback (MVF) [1]. The MVF technique entails placing a mirror perpendicular to the observer’s body midline, creating the illusion of viewing the true limb [2,3]. Although the evidence supporting its effectiveness is still inconclusive [4], several investigations in cohorts of patients with chronic pain or motor deficits demonstrated neuroplasticity processes associated with MVF sensory–motor experience [5,6]. Therefore, MVF-based neurorehabilitation strategies were developed to treat chronic pain conditions such as phantom limb pain (PLP) syndrome [3,7,8], post-stroke hemiparesis [9,10,11], and complex regional pain syndrome [12,13]. These strategies were also successful in facilitating upper-extremities motor recovery [14,15].

Neuroimaging studies in healthy individuals used functional magnetic resonance imaging (fMRI) [16,17,18], near-infrared spectrometry [19,20], and transcranial magnetic stimulation [21,22,23] to investigate the brain neural basis of the effects of the MVF procedure. Results showed an MVF-related increase in cortical excitability (i.e., larger motor-evoked potentials) or activation in premotor, primary motor (M1), and primary somatosensory (S1) cortical areas ipsilateral to the moving hand. Other fMRI and TMS studies extended the focus of the investigation to the whole brain, showing the involvement of the visual and parietal posterior cortical areas—strongly connected with the central sensory–motor regions—as responsible for visuomotor transformation processes occurring during MVF-induced illusory experience [10,18,21,23,24,25,26]. In this regard, an fMRI study investigated the effects of MVF training in low-limb amputees. After 12 sessions of MVF therapy, patients showed a significant reduction of PLP and increased activity in the bilateral orbitofrontal cortex in response to phantom ankle “imaginary” movement [27]. Interestingly, the phantom ankle movements alone induced activation in the premotor and parietal associative cortical areas rather than the primary sensory–motor cortex [27].

The above neuroimaging results had the advantage of a high spatial resolution but were limited in describing the temporal evolution of the MVF-related cortical activation. For this purpose, electroencephalography (EEG) can probe the cortical activity related to motor and sensory events with less spatial resolution but enhanced temporal resolution (milliseconds) [28,29]. Previous EEG investigations measured lateralized related potentials (as an index of M1 activity) during unilateral hand movement tasks performed with MVF illusion. Those potentials showed clear activation of the M1 contralateral to the hand reflected in the mirror and perceived as moving (indeed, immobile) [30,31].

Other EEG studies considered the modulation of the dominant EEG rhythms at 8–12 Hz (i.e., alpha rhythms) recorded in Rolandic central and posterior scalp regions in the resting-state and psychophysically relaxed conditions as a reflection of cortical inhibition of somatomotor and visual–visuospatial cortical regions, respectively [32,33]. These alpha rhythms reduce in amplitude (the so-called event-related alpha desynchronization, alpha ERD) as a sign of cortical activation during the preparation and execution of voluntary movements [32,33,34,35]. Furthermore, central alpha ERD was associated with nociception and sensorimotor interactions (gating) between voluntary hand movements and painful stimuli [35,36].

Previous studies showed consistent alpha ERD (i.e., cortical activation) over frontal and parietal areas during MVF-induced movement illusion [37,38,39]. Abnormal activity in these areas was also linked to dysesthesia (i.e., abnormal limb perceptions associated with pathological pain like PLP) and emotions due to MVF-induced sensorimotor incongruence [40,41,42,43]. Other EEG investigations in healthy participants reported stronger alpha and beta ERD in the bilateral frontal premotor and anterior–posterior cingulate areas when they experienced dysesthesia [42,44] and discomfort sensations [45] induced by sensorimotor incongruence. Notably, dysesthesia is an interesting experimental model as it is characterized by a discomfort like that reported in people with pathological chronic pain [43,46]. Therefore, based on the mentioned investigations in healthy participants, it can be hypothesized that there is a possible neuromodulatory effect of MVF on neurophysiological oscillatory mechanisms generating EEG alpha rhythms in frontal premotor and anterior–posterior cingulate areas.

Keeping in mind the above findings and considerations, we used high-resolution EEG and estimation of alpha ERD cortical sources to shed light on the effects of MVF-induced movement illusion on nociception and frontal premotor and midline cortical areas, including anterior–posterior cingulate and other limbic areas commonly associated with pain processing. These cortical regions may underpin human pain experience related to the affective sense of self, body representation, and interoceptive signals [47,48,49,50,51], dimensions relevant to MVF therapy. Along this line, we tested the novel hypothesis that in healthy adults, the MVF-related illusion of hand movements may affect the nociception and the activation in the premotor and midline cortical areas, as revealed by the alpha ERD. To test this hypothesis, electrical stimuli were applied to the left hand to produce painful or non-painful sensations during unilateral right-hand movements with MVF. As control conditions, the same electrical stimuli were applied during unilateral right-hand and bilateral hand movements without MVF. The alpha ERD was estimated within mathematical sources modeled in the cortical midline and control regions.

## 2. Materials and Methods

### 2.1. Subjects

Thirteen healthy adult male volunteers were recruited for this study (mean age = 26.2, SD = ±4.7). The main exclusion criteria included the presence of chronic pain, neurological diseases, current medical treatment, and participation in other studies involving pain stimulation in the past four weeks. The Edinburgh Handedness Inventory [52] confirmed that all the subjects were right-handed. Remarkably, ten subjects participated in our previous MVF study [39]. Experiments were conducted at the Aalborg University (DK). The Scientific Ethical Committee of Region Nordjylland approved the study (N-20190008) and all the participants signed the informed consent form according to the Declaration of Helsinki.

### 2.2. Experimental Procedure

All participants were seated on a chair with their arms lying symmetrically ahead on a table. Auditory cues of 70 dB, 1000 Hz, and 50 ms of duration [53,54] were used to trigger right index finger movements. These specific auditory parameters were selected to produce an acute short sound, creating a peak in the EEG signal that could be identified and subsequently filtered out. Additionally, previous evidence has demonstrated that this type of passive stimulus does not affect the activity of cortical areas adjacent to the auditory cortex, such as the motor cortex, inferior parietal lobule, and anterior and posterior cingulate cortices [55]. All the subjects underwent three different conditions (80 trials each) in a randomized order. Each condition lasted approx. 13 min, with 10 min breaks between the conditions (the flowchart of the experimental session is represented in Figure 1). The whole session—–including EEG net preparation, training, pain threshold measurement, and experiment—lasted approx. 2.5 h. In the experimental condition (Unilateral with Mirror, or UM+), the mirror was placed on the desk, perpendicular to the subject’s midsagittal plane, with the reflecting face on the right side (Figure 2). Subjects were trained to move their right index finger in response to the auditory cue while watching the image of the reflected moving hand in the mirror to give the illusion of the simultaneous left index finger movement. The position of the left hand behind the mirror corresponded to the image of the left hand reflected in the mirror. This condition was meant to produce sensory–motor interaction between electrical stimuli applied on the left hand and illusory movements of the (mirrored) left hand. In the control conditions, the mirror was removed from the experimental setting and the left hand was directly visible to the participants. Subjects performed the same unilateral right index finger movements in one control condition (Unilateral without Mirror, UM–) in order to see the effects of electrical stimulation on the left hand (i.e., somatosensory cortex on the right hemisphere) without motor interaction. In the other control condition (Bilateral without Mirror, BM–), subjects simultaneously performed movements of both index fingers. This control condition was meant to investigate the effects of the interaction between electrical stimuli and actual movement of the left hand (i.e., sensory–motor cortex of the right hemisphere). Participants were trained to perform a double extension of their index finger with a slow release toward the table (approx. 1 s). All the participants received brief training to perform the movement correctly and keep their left hand as still as possible during the unilateral conditions. In each condition, an electrical stimulus was delivered on the tip of the left index finger 100 ms after the auditory cue to induce cortical sensory–motor interaction (Figure 3). The interval between the stimuli was fixed at 10 s, a sufficient period to reset the desynchronization of the alpha rhythms [35]. A fixed interval was used to allow predictability of the upcoming auditory and electrical stimuli, optimal for studying the anticipatory alpha ERD/ERS responses. However, subjects were not informed of this fixed interval to prevent them from using counting strategies.

### 2.3. Electrical Stimulations and Pain Threshold Detection

Electrical stimulations (5 ms) were provided by a voltage-controlled current stimulator (NoxiSTIM; JNI Biomedical; Aalborg, Denmark). Stimuli were delivered on the tip of the left finger using an electrode with a gold pin cathode and a ring anode (Figure 2). An inactive electrode (sham) was placed on the right index finger to achieve congruent Mirror Visual Feedback (Figure 2). A multichannel data acquisition module (NI-6221, National Instruments Corp., Austin, TX, USA) was used to synchronize the electrical stimuli to the auditory cues.

Subjects were asked to evaluate the intensity of electrical stimuli using a Numerical Scale Rating (NRS) from 0 to 10, where 0 represents the absence of any sensory perception and 10 is the “worst imaginable pain”. Before the pain threshold measurement, 15 stimuli at random intensities were delivered in a familiarization phase. The method of limits was used to determine the pain threshold for each subject. Starting from 0.5 milliamperes (mA), the current intensity was progressively increased until a rating of 5 out of 10, namely, the minimum current intensity producing a painful sensation [56]. With this procedure, we aimed to reach a fixed intensity that was perceived half the time as painful (NRS > 5) or non-painful (NRS < 5). The same fixed individual pain threshold was used during the three conditions (mean mA = 5.5; SD = ± 3.1). In between the trials (approx. 3–4 s after the cue), subjects were asked to provide a numerical value (decimals included) of the average pain intensity of the last 5 trials. Subjects vocally rated the stimuli intensity, and each value was noted by the experimenter on an Excel spreadsheet. The speaking artefacts were removed in the offline analysis. Since each condition consisted of 80 trials, 16 values per condition were considered in the analysis. Due to the fluctuant subjective perception of fixed stimuli, painful (NRS > 5) and non-painful (NRS < 5) responses were obtained for each condition. If one condition did not show fluctuation in the perceived pain intensity, the single dataset was removed from the analysis. Blocks with fluctuation were defined as blocks presenting a ratio of at least 30%/70% of painful/non-painful trials or vice versa. Moreover, to avoid any sort of habituation or sensitization phenomena, we excluded those blocks that did not show a random distribution between painful and non-painful trials (see RUNS test in Control Analysis and Appendix A). The final samples considered in the statistical analysis were UM+ = 8 subjects, UM– = 13 subjects, and BM– = 12 subjects. Of interest, the painful epochs (NRS > 5) were 36.5% in the UM– condition, 58.5% in the BM– condition, and 53.9% in the UM+ condition.

### 2.4. EEG Recording and Preprocessing

Scalp EEG activity was recorded using an active 64-channel system (g.HIamp amplifier, g.tec medical engineering GmbH, Graz, Austria) following the 10–10 international system. Ground and reference electrodes were positioned on the forehead and ear lobes, respectively. Electrode impedance was maintained under 5 kΩ, and data were sampled at 1200 Hz. Eye movements and blinks were detected using Fp1 and Fp2 electrodes and subsequently removed during the offline analysis. Data preprocessing was conducted using the EEGLAB 2021.1 [57] freeware toolbox in MatLab R2018b (MathWorks, Inc., Natick, MA, USA). Following visual inspection, data were filtered with a zero-phase basic FIR filter (passband edges: 0.3–40 Hz, cutoff frequencies: −6 dB, filter order: 846, transition band: 1 Hz), resampled to 256 Hz, and re-referenced to the average of all channels to mitigate reference-related effects from online recording [58]. Data were then segmented into 80 epochs of 9 s each. Artefact removal involved identifying and eliminating artefacts such as blinks, muscle movements, and 50 Hz noise using the Infomax independent component analysis (ICA) algorithm [59]. Approximately 7.9% of artifact-laden epochs and 7.3% of artefact-related components were excluded from the EEG datasets. Subsequently, global EEG spectra were computed using FFT-based methods to determine the individual alpha frequency peak (IAFp). IAFp was defined as the frequency exhibiting the highest peak within the alpha band (8–12 Hz) [60]. EEG data were filtered at the individual alpha frequency band range from IAF–2 Hz to IAF + 2 Hz (mean IAFp = 10.1; SD = ± 1.0) [60,61].

### 2.5. Cortical Sources Estimation

The freeware “exact Low-Resolution Brain Electromagnetic Tomography” (eLORETA) [62] was utilized for cortical sources estimation using preprocessed EEG data as input. The eLORETA freeware is a functional imaging technique that belongs to a group of standardized linear inverse solution procedures, modelling the 3D distributions of EEG sources within a head volume conductor model, including the scalp, skull, and brain [63,64]. eLORETA addresses the EEG inverse problem by estimating the current density values at the voxel level, offering a brain source space limited to the cortical grey matter, consisting of 6239 voxels (5 mm resolution, with each voxel containing an equivalent current dipole), based on the Montreal Neurological Institute template (MNI152). For each voxel, eLORETA provides information about the MNI coordinates, the lobe, and the Brodmann area (BA). The input for the source estimation is a spectral power density computed by 62 scalp electrodes, as eLORETA source analysis is a free-reference method [65]. Due to the low spatial resolution characteristic of the EEG techniques, adjacent Brodmann areas were clustered. The clusters included BAs 1–2–3 together with BA 4 (sensory–motor area), BA 24 together with BA 32 (anterior cingulate cortex), BA 44 with BA 45 (inferior frontal gyrus), and BA 9 with BA 46 (superior frontal gyrus or dorsolateral prefrontal cortex).

### 2.6. Alpha ERD/ERS Calculation

Standard quantification of the alpha ERD/ERS was conducted for both the anticipation and execution stages of the movements [32,36,66,67,68]. The alpha ERD/ERS was calculated using the formula: ERD/ERS% = (E − R)/R × 100, where E represents the power density at the “Event” period (1 s) and R represents the power density at the “Rest” periods (1 s). The “Rest” period was defined as the interval from 5 s to 4 s before the auditory cues that triggered the movement. The anticipation “Event” was defined as the period of 1 s before the auditory cues. Finally, the execution “Event” was defined as the 1 s period from 250 ms to 1250 ms after the auditory cues, with the alpha ERD peak within this interval considered in the statistical analysis. The first 250 ms post-cue were excluded to remove auditory-evoked potentials (N1-P2 complex) [69] and pain-evoked potentials (N2-P2 complex) [35,70,71]. Negative percentage values represented the alpha ERD, indicating cortical activity, while positive values represented the alpha ERS, indicating cortical inhibition [32,66]. For scalp analysis, electrodes were clustered for each hemisphere at the frontal (F3 and FC3; F4 and FC4) and centro–parietal (C3, CP3, and P3; C4, CP4, and P4) levels. The eLORETA solutions were utilized as input to compute the cortical voxel-level ERD/ERS in the alpha frequency band.

### 2.7. Statistical Analysis for the ERD/ERS Scalp Distribution

Linear mixed models for repeated measures were used to assess whether the subjective perception of the stimulus intensity depended on the MVF. The fixed factors were Condition (UM+, UM–, and BM–) and Stimulus (pain and no-pain), whereas the random factor was the number of subjects in each group. The models were fitted separately in the left and right hemispheres for the frontal (F3-FC3 and F4-FC4) and centro–parietal (C3-CP3-P3 and C4-CP4-P4) clusters of electrodes, as well as for the anticipation and execution phases of the event. The linear mixed model statistical analysis was carried out using IBM SPSS Statistics 27 (IBM, New York, NY, USA).

### 2.8. Statistical Analysis for the Alpha ERD/ERS Cortical Sources

To investigate differences in cortical activation between stimuli perceived as painful and non-painful, statistical nonparametric mapping voxel-based wise randomization tests (5000 permutations) were employed for the alpha ERD/ERS eLORETA solutions for each condition (UM+, UM–, and BM–) and stage of the event (Anticipation and Execution). The results derived from this statistical approach are equivalent to those obtained using a statistical parametric method with multiple comparison corrections [72]. This nonparametric permutation approach is appropriate for analyzing low sample sizes with low degrees of freedom [72,73]. The results are shown as voxel-based T statistics maps with corrected *p* < 0.05 (randomization tests correction) [72]. To reduce the type II error (false negatives), an uncorrected significance threshold of *p* < 0.001 was employed. Only cortical areas comprising at least 5 voxels were included in the analysis. Statistical analyses were conducted using the eLORETA statistical toolbox [62]. The significance of the results was statistically confirmed by the effect size (r) for the two-sided Wilcoxon signed rank test, a nonparametric test used to compare paired groups [74,75]. In particular, Wilcoxon’s r measured the effect size of the difference between Pain and No pain in the ERS/ERD% corresponding to each’s most significant voxel (Table 1). The effect size was calculated for each condition (UM–, BM–, and UM+) and movement stage (ANT and EXE). Wilcoxon’s r values range from −1 to 1 and are interpreted as |0|-|0.1| = no effect or very small effect, |0.1|-|0.3| = small effect, |0.3|-|0.5| = medium effect, and |0.5|-|1| = large effect [76]. Wilcoxon’s r values are reported in the extreme right column in Table 1. The effect size was calculated using the *effsize* package for RStudio (v. 4.3.1).

### 2.9. Control Analysis

Event-related potentials (ERPs) in response to the auditory cues and electrical stimuli were extracted from the Cz electrode. Successively, the waveforms resulting from painful and non-painful blocks were averaged across the subjects for each condition and paired samples *t*-tests were used for statistical comparison. The ERPs considered were the N1-P2 (auditory), P2-N2 complex (electrical stimulus), and P3 (attentional). Furthermore, the randomness of the painful and non-painful epochs was tested by performing the RUNS test for each subject and condition. Finally, NRS values were compared to control differences in the subjective perception of the stimulus intensity for each condition. All the statistical analyses were performed using IBM SPSS Statistics 27 (IBM, New York, NY, USA). Furthermore, time–frequency maps for C3 and C4 electrodes were plotted to provide an estimation of the whole frequency spectrum (0–40 Hz) for each condition and movement stage (Appendix A). The results from the control analyses are reported in detail in Appendix A.

## 3. Results

### 3.1. Scalp Topography

Figure 4 shows the scalp distribution of the alpha ERD/ERS for the three conditions (UM–, BM–, and UM+) at the anticipation (1000 ms before the auditory cue) and execution (peak within 1250 ms after the auditory cue) of the event. The cortical activity patterns resulting from the blocks perceived as painful (NRS > 5) and non-painful (NRS < 5) are thus compared. In the control conditions without MVF (UM– and BM–), results indicate a similar centrally distributed alpha ERD in the anticipation phase of the event. When considering the execution phase, the alpha ERD is more widespread in the frontal–central areas in the non-painful than painful blocks in both conditions. In the experimental MVF condition, the maps show a slightly stronger alpha ERD in the painful than non-painful blocks for both the anticipation and execution phases of the movement. Moreover, the execution phase shows a prominent parietal–central distribution in the painful rather than non-painful blocks.

However, the linear mixed model analyses performed to assess the interaction between the stimulus intensities and the conditions did not indicate any statistically significant effect for the frontal and central brain regions (*p* > 0.05).

### 3.2. Control Conditions without MVF

This section of the results considers the differences between painful and non-painful blocks for the control conditions without the MVF illusion (i.e., UM– and BM–). Figure 5 and Figure 6 display the spatial distribution of *p* values from the parametric statistical maps (Student’s *t*-tests) for the eLORETA solutions of the alpha ERD/ERS, shown in the first and second rows. Table 1 lists the significant T values along with the voxel cluster sizes and their locations.

For both conditions, the anticipation and execution phases of the event are characterized by a stronger activation (alpha ERD) in the non-painful than painful blocks. In particular, the UM– condition shows this pattern in the left anterior cingulate (BA 32; peak value: T = 4.36; *p* < 0.05 corrected; r = −0.87) and inferior parietal cortex (BA 40; peak value: T = 3.95; *p* < 0.05 corrected; r = −0.81) in the anticipatory phase, and no significant differences in the execution phase. In the BM– conditions, the differences appear in the bilateral sensory–motor (BAs 1–2–3, 4; peak value: 4.61; *p* < 0.05 corrected; r = −0.84), inferior parietal (BA 40; peak value: T = 4.46; *p* < 0.05; r = −0.89) cortical areas, and bilateral inferior frontal gyrus (BAs 44–45; peak value: T = 3.36; *p* < 0.001 uncorrected; r = −0.81), for the anticipation phase (Figure 5). Furthermore, the execution phase of the event (Figure 6) shows those differences in the right superior (BAs 9–46; peak value: T = 4.11; *p* < 0.05 corrected; r = −0.85) and inferior (BAs 44–45; peak value: T = 3.79; *p* < 0.001 uncorrected; r = −0.90) frontal gyri, as well as right anterior cingulate cortex (BA 32; peak value: T = 3.67; *p* < 0.001 uncorrected; r = −0.81).

### 3.3. Experimental Condition with MVF

The voxel-by-voxel comparison results between painful and non-painful blocks in the experimental MVF (UM+) condition are hereby reported. The spatial distribution of the *p* values for the Student’s *t*-tests is illustrated in Figure 5 and Figure 6 (bottom rows). The significant T values, as well as the voxel cluster size and localizations, are summarized in Table 1.

Unlike the control conditions, in the UM+ condition, the alpha ERD observed in the painful blocks was stronger than the alpha ERD in the non-painful blocks. During the anticipation phase (Figure 5), the main differences are observed in the right posterior cingulate (BA 30; peak value: T = −4.16; *p* < 0.05 corrected; r = 0.94) and the inferior posterior parietal cortex (BA 40; peak value: T = −4.73; *p* < 0.05 corrected; r = 0.93). During the execution phase (Figure 6), these differences are widespread in the frontal and posterior areas. Results showed significant differences in the right polar frontal (BA 10; peak value: T = −4.31; *p* < 0.05 corrected; r = 0.94) and inferior posterior parietal (BA 40; peal value: T = −4.67; *p* < 0.05 corrected; r = 0.86) cortical areas. The same was true in the right middle temporal gyrus (BA 39; peak value: T = −4.75; *p* < 0.05 corrected; r = 0.93) and precuneus (BA 19; peak value: T = −3.87; *p* < 0.001 uncorrected; r = 0.84), as well as in the bilateral anterior cingulate (BAs 24–32; peak value: T = −4.66; *p* < 0.05 corrected; r = 0.85).

## 4. Discussion

In the present exploratory study, we hypothesized that the MVF-induced illusion of hand movements may affect the nociception and the activation in those cortical midline areas associated with pain processing. For this purpose, we used an original methodological approach. Electrical stimuli at about the pain threshold were applied to the left hand to produce painful or non-painful sensations during unilateral right-hand movements with the MVF and, as control conditions, during unilateral right and bilateral hand movements without the MVF. The eLORETA freeware estimated the EEG source activity, and the alpha ERD (as a sign of cortical activation) reflected the neurophysiological oscillatory mechanisms underlying the painful and non-painful sensations associated with the MVF-related movement illusion.

The results of a control analysis showed that the vertex N1-P2 and/or N2 peaks of the sensory-evoked potentials exhibited higher amplitude in relation to the painful over the non-painful sensations during both MVF and noMVF conditions (Appendix A). These results corroborate the reliability of the participants’ subjective sensations, in line with previous studies demonstrating that those sensory-evoked potentials can be considered trustworthy neurophysiological signatures of enhanced cortical arousal and pain experience in humans [69,70,71].

The core results of the present study showed that, compared to the painful sensations, the non-painful sensations during the MVF condition were specifically characterized by lower alpha ERD (cortical activation) estimated in the right cortical midline regions (i.e., medial prefrontal, anterior and posterior cingulate, parietal cortex, and precuneus), angular gyrus, and inferior parietal lobule contralateral to the electric hand stimulation and ipsilateral to the hand movements. In contrast, the non-painful sensations during the control noMVF conditions were specifically characterized by stronger alpha ERD estimated in the left lateral prefrontal and inferior parietal lobule regions ipsilateral to the electric hand stimulation and contralateral to the hand movements. Large effect sizes (i.e., r > 0.8) confirmed the significance of these findings. Although the results of this study are preliminary and need to be interpreted with caution, a functional interpretation is provided in the following section.

### 4.1. Effects of MVF-Induced Movement Illusion on Cortical Activity and Pain

During the MVF condition, the non-painful over painful sensations were specifically related to lower alpha ERD in the right cortical midline regions of the limbic and default mode network, in the right angular gyrus of the default mode network, and in the right inferior parietal lobule of the ventral attention network. In contrast, the non-painful sensations in the noMVF conditions were characterized by higher alpha ERD in the left lateral prefrontal and inferior parietal lobule of the ventral attention networks.

The current findings emphasize the role of alpha frequency neurophysiological oscillatory mechanisms in modulating the deactivation within cortical regions pertinent to human nociception. These results complement previous neuroimaging research in humans, which highlighted: (1) the involvement of medial prefrontal and anterior–posterior cingulate cortices in processing the affective aspect of pain [77,78]; (2) the activation of cortical midline regions in relation to the unpleasant sensations elicited by painful stimuli [79,80,81]; and (3) the activation of a posterior cortical subnetwork including the inferior parietal lobe, posterior cingulate cortex, and precuneus, associated with affective and cognitive components of the pain experience, such as pain anticipation and self-awareness [82]. Additionally, the present findings align with previous evidence indicating that sensorimotor incongruence in healthy individuals can lead to alterations in limb perceptions and emotional responses (e.g., dysesthesia) [40,42,45]. These studies have linked dysesthesia to abnormal EEG alpha and beta rhythms observed in parietal, midline premotor, and cingulate cortical areas [44,45]. Notably, such discomfort is similar to that reported in people with pathological chronic pain [43,46].

Although the present findings are preliminary and must be interpreted with caution, they offer—along with the previous neuroimaging results—intriguing insights into the neurophysiological mechanism underlying the experience of pain during MVF illusion. In the current MVF condition, the decreased alpha ERD observed in the right cortical midline limbic regions may be linked to non-painful sensations that diminish the affective sense of self and internal body perception [42,45,47,50,51]. Furthermore, the reduced alpha ERD in the right default mode network regions could be associated with weakened internal representations related to bodily sensations (interoception) [83,84,85]. Conversely, in the noMVF conditions, the heightened alpha ERD in the left lateral prefrontal and inferior parietal lobule may be associated with non-painful sensations related to significant attentional allocation and cognitive states towards the actual moving hand(s) for “body gnosis”, affecting neural signals induced by electrical hand stimulation [86,87]. Unfortunately, EEG techniques do not have the fine spatial resolution necessary to disentangle the different subcomponents of a given midline cortical region belonging to either the limbic or default mode network.

During the present MVF condition, the participants may experience the self-oriented attentional and cognitive states associated with the illusory sense of ownership (“the illusory moving hand belongs to my body”) and agency (“I am moving the hand reflected in the mirror”) for the moving hand reflected in the mirror. According to previous alpha ERD evidence from our group [39,88], these states may be related to the activation of lateral prefrontal, premotor, and parietal cortical regions in the hemisphere ipsilateral to the true unilateral hand movements and contralateral to the moving hand reflected in the mirror. Notably, in the present experiments, the left inferior parietal lobule was the only region of that cortical network showing an effect related to pain. Specifically, the non-painful (rather than painful) sensations during the MVF condition were associated with lower alpha ERD in the right inferior parietal lobule, as opposed to the higher alpha ERD in the left inferior parietal lobule during the noMVF conditions. It can be speculated that during the MVF condition, the non-painful sensations were also related to a low activation in the right lateral parietal area integrating the visual–somato(nociceptive)–motor information regarding the electrically stimulated left hand.

### 4.2. Clinical Implications and Future Perspectives

The 64-channel EEG system used in the present study provided a precise delineation of the temporal and spatial evolution of the alpha oscillatory mechanisms during movement preparation and execution, as well as sensory stimuli expectancy and conscious experience during the MVF procedure. Specifically, the present findings suggest that the EEG alpha rhythms may be insightful markers of the neurophysiological underpinning of nociception and sensorimotor interactions during MVF-related rehabilitative processes, such as motor relearning [89,90,91] and enhanced transcallosal communication [92,93]. If future studies confirm the present results in patients with chronic pain undergoing MVF therapy, even at the individual level, the assessment of the alpha ERD may serve as a neurophysiological biomarker for restored sensory–motor information processing or tracking patients’ motor recovery trajectory. An additional aspect of the present findings—potentially relevant in the clinical context—is the strong relationship between nociceptive experience and attentional and internal body representation during MVF experience. The EEG alpha rhythms may be used in future studies as a biomarker indicating the best MVF settings to induce the modulation of EEG alpha source activity in the limbic system. Along this line, the MVF settings using virtual reality technologies may be especially promising [94,95]. Future neurorehabilitation methods may leverage MVF-based virtual reality technologies to provide greater illusory immersion and enhance the range of movement, including asymmetric movements that are otherwise impossible with traditional MVF methods.

Future multimodal EEG–fMRI studies with higher spatial resolution should shed light on the effects of the MVF-related movement illusion on the nociception and the functional connectivity between all cortical and subcortical (e.g., amygdala–hypothalamus, amygdala–septum, hippocampus–mammillary bodies, etc.) components of the limbic system in both healthy participants and patients with chronic pain. A modulation in this connectivity is expected to impact the integration of primary needs, the emotional value of the painful stimuli, and interoceptive signals related to the person’s experience in relation to the wellness–illness axis during painful stimulations [48,49,50]. Furthermore, future studies in both healthy participants and patients with chronic pain may investigate the functional connectivity between the limbic and the default mode networks in connection with the MVF effects on nociception and chronic pain. These studies may suggest new pathophysiological, neurophysiological, and functional neuroanatomical targets and companion biomarkers to integrate MVF therapy and pharmacological analgesic treatments.

### 4.3. Study Limitations

Due to its exploratory nature, this study has a few significant limitations.

Firstly, this study used a small population of healthy male volunteers to preliminarily test the working hypothesis. As a possible effect of this limited participants’ sample, the participants did not clearly exhibit the expected fluctuations in perceived pain intensity. Consequently, the three groups presented different sample sizes, necessitating the use of a more complex mixed-model statistical design. At this early stage of the research, we enrolled only healthy male participants to avoid the potential influence of hormonal fluctuations during the menstrual cycle on pain perception [96]. The present results encourage the planning of future validation studies in a larger sample of participants, including both males and females (taking into account the menstrual cycle in the latter).

Secondly, the phasic electrical stimulations used in this study allowed us to investigate fundamental neurophysiological mechanisms underpinning human nociception and sensorimotor interactions during the MVF experience. The present results encourage the application of the present methodological approach in healthy participants using the model of tonic painful stimulations, which is more relevant in relation to chronic pain and patients with chronic pain conditions typically undergoing MVF-based therapies.

Thirdly, the intrinsic limitations of the EEG technique should be considered in the interpretation of the present results. Although the cortical midline structures present dense connections with the limbic system (e.g., amygdala, thalamus, hippocampus), the involvement of specific subregions of those brain structures could not be probed due to the intrinsically limited spatial resolution of EEG source solutions, even with the present high-resolution techniques (e.g., centimeters).

## 5. Conclusions

In this exploratory study, we hypothesized that in healthy adults, the MVF-induced movement illusion could alter midline cortical activation, as indicated by the EEG alpha neurophysiological oscillatory mechanisms. In comparison to painful sensations, non-painful sensations were linked to lower alpha ERD in the cortical midline, angular gyrus, and lateral parietal regions during the MVF experimental condition and higher alpha ERD in the lateral prefrontal and parietal regions during the MVF-free control conditions. These preliminary observations suggest that the MVF-induced illusion might influence nociception and neurophysiological mechanisms by reducing activation in cortical limbic and default mode regions, potentially affecting the sense of self, internal body perception, and attention to electrical signals from the stimulated hand. Although preliminary, these findings deserve attention and further verification is essential to develop beneficial approaches in those patients with chronic pain and motor impairments. Future research should use larger samples of healthy participants and patients with chronic pain to achieve generalizable results on both the cortical neurophysiological model of nociception and sensorimotor interactions and MVF neuromodulatory effects on the limbic system in the therapy of patients with chronic pain.

## Figures and Tables

**Figure 1 brainsci-14-00696-f001:**
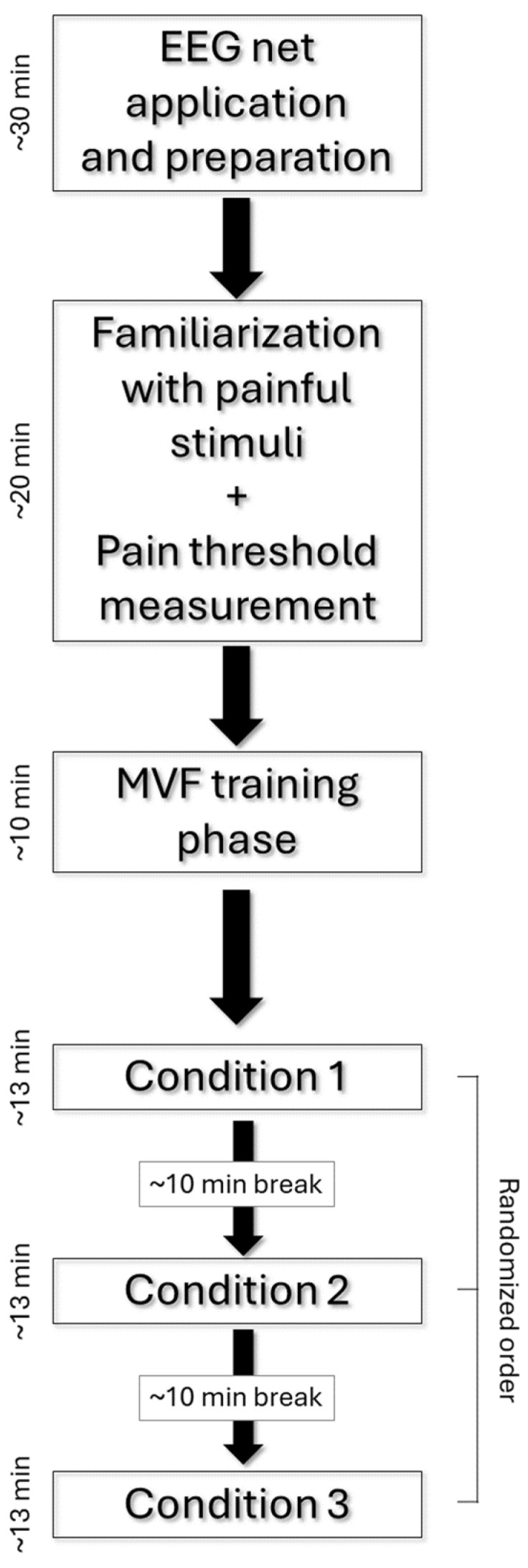
Experimental session flowchart.

**Figure 2 brainsci-14-00696-f002:**
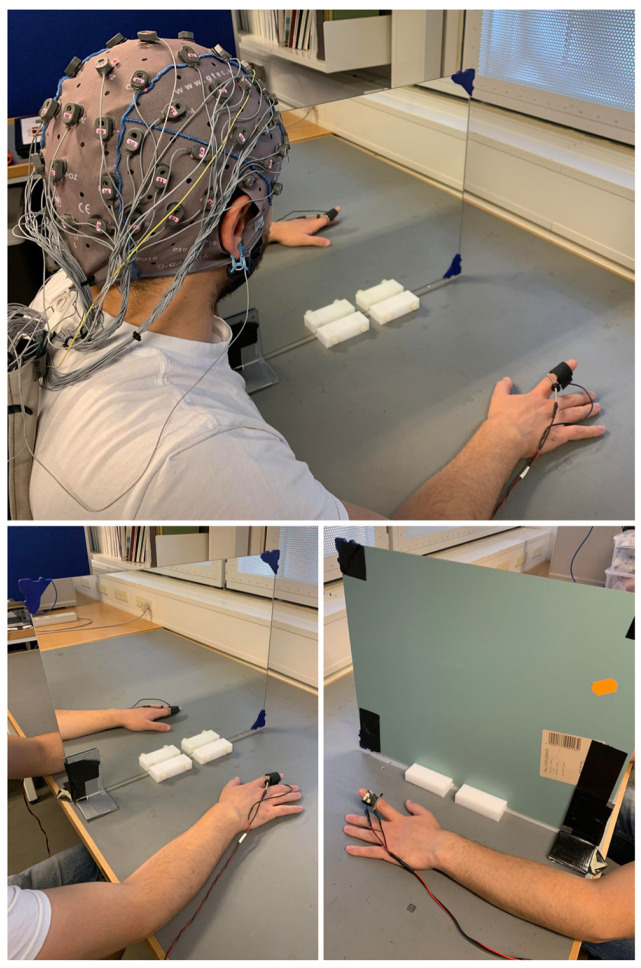
Experimental setup during the Mirror Visual Feedback (MVF) procedure (Unilateral with Mirror or UM+ condition). The mirror is placed in the subject’s midsagittal plane to give the illusion of ownership of the left hand. An electrode delivering the stimulations was placed on the index finger of the left hand (bottom right panel). A sham electrode was placed on the right index finger to strengthen the illusory feeling through congruent visual feedback. Movements consisted of double extension (1 s approx.) with a slow release toward the table.

**Figure 3 brainsci-14-00696-f003:**
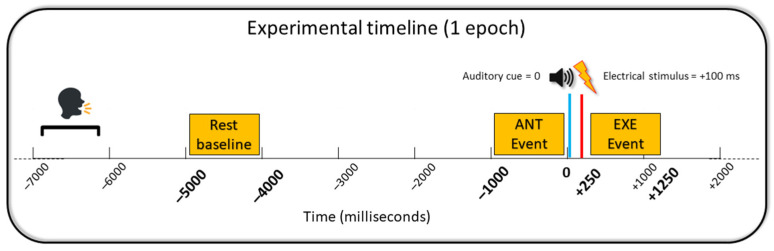
The above figure illustrates the sequence of one trial of the experimental paradigm. In the experiment, a fixed 10 s interstimulus interval was applied. During the offline analysis, 9 s epochs were extracted (from 7 s before the auditory cue to 2 s after the cue). In the above figure, the time is expressed in milliseconds (ms). The 0 (zero) time corresponds to the auditory cue triggering the right index finger movement (blue line). In each trial, electrical stimuli were delivered on the left index finger after 100 ms from the auditory cue (red line). In between the trials, subjects were asked to vocally rate the sensation related to the electrical stimulation on a scale from 0 to 10 (painful > 5; non-painful < 5). The anticipation (ANT) event period was defined as the time interval of 1000 ms before the 0, whereas the execution (EXE) period was defined as the time interval between 250 and 1250 ms after the 0 (yellow boxes). The alpha event-related de/synchronization (ERD/ERS) was calculated for the ANT and EXE periods in relation to a baseline period represented by the yellow box between 5000 and 4000 ms before the 0 (yellow box).

**Figure 4 brainsci-14-00696-f004:**
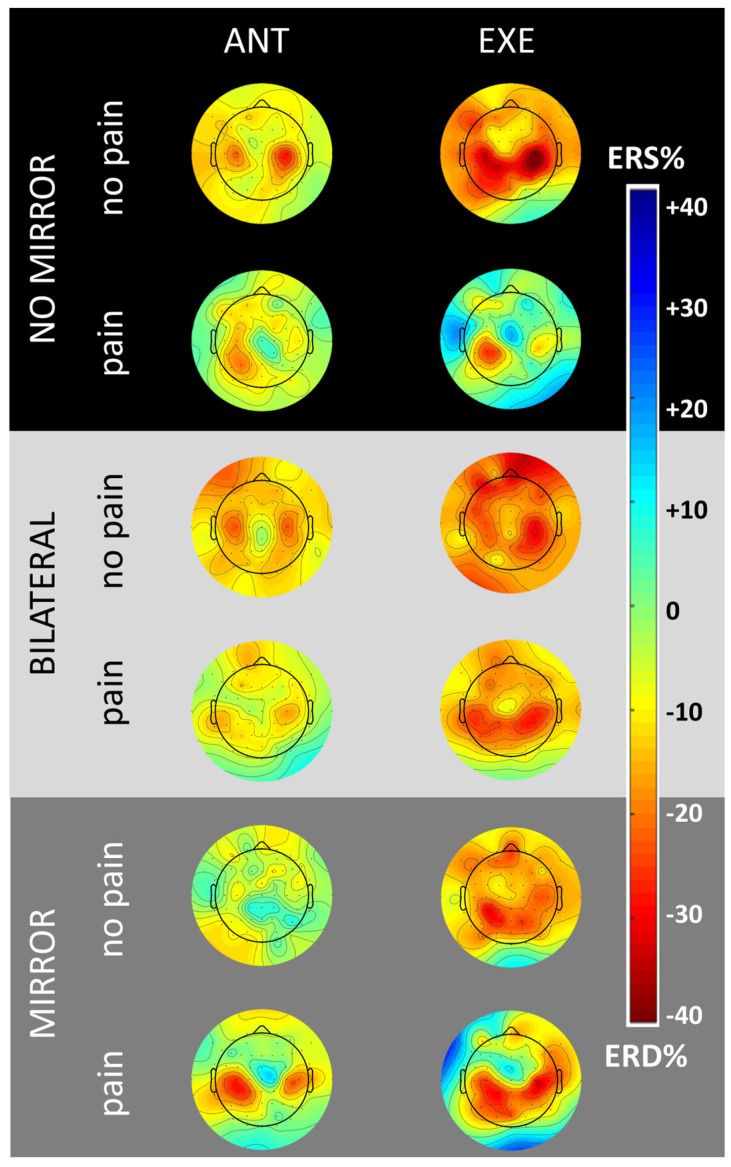
Across subjects’ mean 2D maps of the alpha ERD/ERS distribution over the scalp for each condition (Unilateral without Mirror, Bilateral without Mirror, and Unilateral with Mirror), phase of the event (Anticipation and Execution), and subjective perceived intensity of the stimulus (pain, no-pain). In the maps, the alpha event-related desynchronization (ERD) and synchronization (ERS) are represented with red and blue colors, respectively. For the anticipation phase (ANT), the 1 s interval before the auditory cue is reported as a scalp map. For the execution phase (EXE), the scalp maps are represented by 250 ms time windows after the auditory cue and the interval showing the highest peak of the alpha ERD was reported in the figure. The maps indicate a centrally distributed alpha ERD in the ANT phase and frontocentral alpha ERD in the EXE phase during the two control conditions for both pain and non-pain blocks. In the experimental mirror condition, the maps show a stronger but not statistically significant alpha ERD at the central level in the painful over the non-painful blocks for both stages of the events (ANT and EXE).

**Figure 5 brainsci-14-00696-f005:**
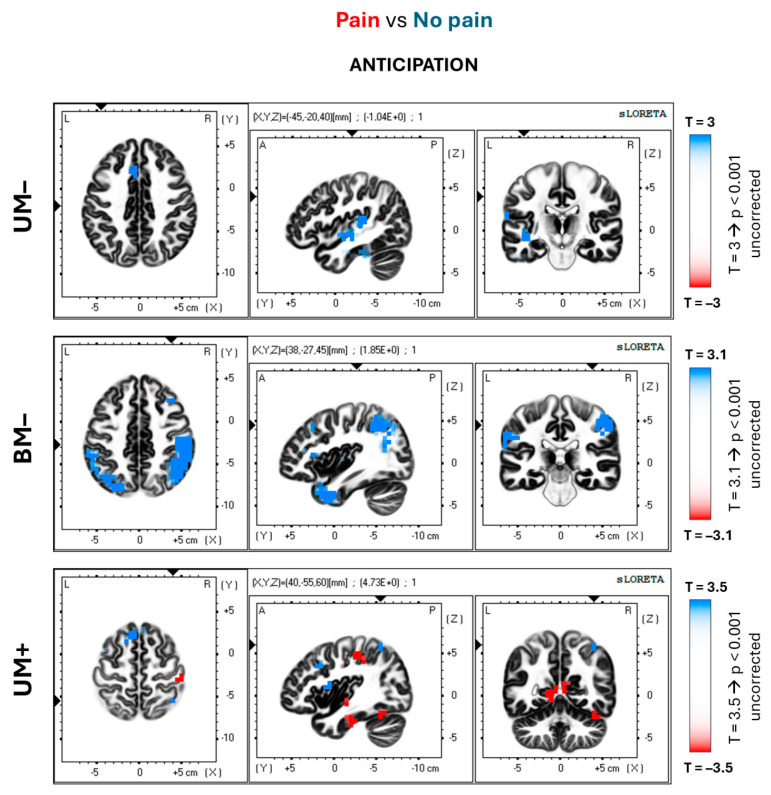
Spatial distribution of the voxel-by-voxel significant *p* values relative to the Student’s *t*-test for the alpha ERD/ERS eLORETA solutions. The above figure illustrates the comparisons between pain and no-pain blocks for each condition (UM–, BM–, and UM+) at the anticipation phase of the event. The axial, sagittal, and coronal sections are represented. The T values corresponding to the uncorrected significance threshold of *p* < 0.001 are shown on the right side for each condition (the T values change as the sample sizes are different among the groups). In the maps, the red voxels show the areas where the alpha ERD is significantly stronger in the painful than the non-painful blocks (posterior cingulate and inferior parietal lobule). Conversely, the blue voxels show the areas where the alpha ERD is significantly stronger in the non-painful than the painful blocks (sensory–motor and parietal associative cortical areas).

**Figure 6 brainsci-14-00696-f006:**
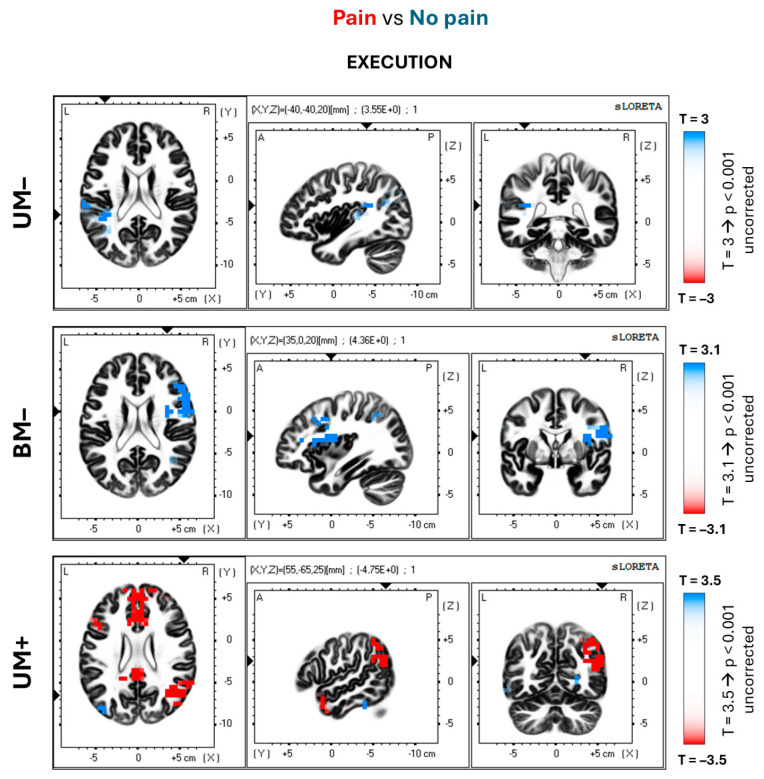
Spatial distribution of the voxel-by-voxel significant *p* values relative to the Student’s *t*-test for the alpha ERD/ERS eLORETA solutions. The figure illustrates the comparisons between the painful and non-painful blocks for each condition (UM–, BM–, and UM+) at the execution phase of the event. The axial, sagittal, and coronal sections are represented. The T values corresponding to the uncorrected significance threshold of *p* < 0.001 are shown on the right side for each condition (the T values change as the sample sizes are different among the groups). In the maps, the red voxels show the areas where the alpha ERD is significantly stronger in the painful than the non-painful blocks (cortical midline structures of the limbic system, default mode network, and attentional network). Conversely, the blue voxels show the areas where the alpha ERD is significantly stronger in the non-painful than the painful blocks (sensory–motor and parietal associative cortical areas).

**Table 1 brainsci-14-00696-t001:** Voxel-based Student’s *t*-test analysis for the eLORETA sources of the alpha ERD at the Brodmann areas (BAs) level. ** Significance threshold of *p* < 0.05 corrected for multiple comparisons; * significance threshold of *p* < 0.001 uncorrected; n.s. not significant. The cluster size denotes the number of voxels that meet the significance threshold for multiple comparisons (uncorrected threshold in brackets). For the most statistically significant voxels, its Montreal Neurological Institute (MNI) coordinates, T value, and effect size (Wilcoxon’s r) are reported. Effect size values range from −1 to 1 and were interpreted as |0|-|0.1| = no effect or very small effect, |0.1|-|0.3| = small effect, |0.3|-|0.5| = medium effect, and |0.5|-|1| = large effect. For a better comprehension of the results, it must be noted that the alpha ERD represents a negative percentage indicating decreased power density during the event compared to the baseline. Therefore, when comparing painful versus non-painful epochs, negative T values indicate higher alpha ERD during painful epochs, while positive T values indicate higher alpha ERD during non-painful epochs. In the two control conditions (UM– and BM–), both the anticipation and execution phases of the event are characterized by stronger alpha ERD in the non-painful than painful blocks (as reflected by positive T values) in the sensory–motor and parietal associative areas. In the experimental condition (UM+), the anticipation and execution phases of the event are characterized by stronger alpha ERD in the painful rather than non-painful blocks (negative T values) in those cortical midline regions of the limbic system, default mode network, and attentional network.

ANTICIPATION									
Conditions Comparisons	BAs	Cluster Size	Region	Hemisphere	MNI Coordinates			T Value	Wilcoxon’s r
					x	y	z		
UM–	32	5 (9)	Anterior cingulate	L	−5	20	40	4.36 **	−0.87
(Pain vs. No pain)	40	3 (8)	Inferior parietal lobule	L	−65	−25	20	3.95 **	−0.81
BM– (Pain vs. No pain)	1–2–3, 4	13 (36)	Central gyrus	L/R	60	−30	45	4.61 **	−0.84
	40	113	Inferior parietal lobule	L/R	−65	−40	35	4.46 **	−0.89
	44–45	15	Inferior frontal gyrus	L/R	−60	5	15	3.36 *	−0.81
UM+	30	2 (8)	Posterior cingulate	R	5	−55	5	−4.16 **	0.94
(Pain vs. No pain)	40	1 (5)	Inferior parietal lobule	R	40	−55	60	−4.73 **	0.93
**EXECUTION**									
**Conditions Comparisons**	**BAs**	**Cluster Size**	**Region**	**Hemisphere**	**MNI Coordinates**			**T Value**	**Wilcoxon’s r**
					**x**	**y**	**z**		
UM– (Pain vs. No pain)	40	5	Inferior parietal lobule	L	−55	−25	15	3.34 ^n.s.^	−0.75
BM– (Pain vs. No pain)	9–46	7 (21)	Superior frontal gyrus	R	55	15	30	4.11 **	−0.85
	32	7	Anterior cingulate	R	15	45	−5	3.67 *	−0.81
	44–45	35	Inferior frontal gyrus	R	60	15	15	3.79 *	−0.9
UM+ (Pain vs. No pain)	10	2 (19)	Medial frontal gyrus	R	10	50	15	−4.31 **	0.94
	19	5	Precuneus	R	30	−75	35	−3.87 *	0.84
	24–32	2 (11)	Anterior cingulate	L/R	5	25	15	−4.66 **	0.85
	39	2 (17)	Angular gyrus	R	55	−65	25	−4.75 **	0.93
	40	3 (10)	Inferior parietal lobule	R	45	−55	45	−4.67 **	0.86

## Data Availability

The EEG datasets have been reposited on Figshare at the following site: https://doi.org/10.6084/m9.figshare.24851748.v1 (accessed on 18 December 2023).

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
