# Peer review of "Neurophysiological Oscillatory Mechanisms Underlying the Effect of Mirror Visual Feedback-Induced Illusion of Hand Movements on Nociception and Cortical Activation"

_brainsci, 2024, doi:10.3390/brainsci14070696_

Round 1

Reviewer 1 Report

Comments and Suggestions for Authors

Introduction: This section needs to be detailed description of this overall study and highlight objectives of this research in the end. The literature review could be updated to include more recent studies that discuss the neurophysiological implications of MVF.

Materials and Methods: The methodology is detailed but has significant limitations:

  1. Sample Size and Composition: The sample size is small and limited to healthy adult males. This limits the generalizability of the findings. Future studies should consider a larger and more diverse sample, including females and individuals with relevant clinical conditions.
  2. Control Conditions: While the experimental design includes control conditions, the description of these controls and their justification could be clearer. It is crucial to elaborate on how these controls adequately address potential confounding variables.
  3. Equipment and Measurement Techniques: The description of EEG equipment and settings is adequate, but the justification for the chosen parameters, particularly the EEG frequency bands analyzed, could be strengthened.

Results: The results are clearly presented, but the interpretation is cautious and somewhat limited:

  1. Statistical Power: Given the small sample size, the statistical power of the study is questionable.
  2. Data Presentation: Some of the figures and tables are overly complex. Simplifying these or providing more detailed captions could help in better communicating the findings.

Discussion: The discussion provides a good link between the findings and existing literature, but it can be improved in several ways:

  1. Broader Implications: Expand on the clinical implications, suggesting specific changes or adaptations in clinical practices or future research.
  2. Limitations: The discussion of limitations is somewhat superficial. Detailed limitations, including the EEG's spatial resolution and the study's validity need to be added in this section.

Conclusions:The conclusions are appropriately cautious given the exploratory nature of the study. However, they could more strongly emphasize the need for further research in more clinically diverse populations

Comments on the Quality of English Language

Minor English language editing is recommended. 

Author Response

REVIEWER 1

Introduction: This section needs to be detailed description of this overall study and highlight objectives of this research in the end. The literature review could be updated to include more recent studies that discuss the neurophysiological implications of MVF.

Authors: We thank Reviewer 1 for the constructive and punctual comments. The Introduction has been thoroughly revised to provide a clearer explanation of the studies' objectives in neurophysiology and pain processing, as well as the cortical areas involved in analgesic processes and affective aspects of pain. Given that the Introduction was entirely rephrased, we did not highlight this part in yellow.

Materials and Methods: The methodology is detailed but has significant limitations:

  1. Sample Size and Composition: The sample size is small and limited to healthy adult males. This limits the generalizability of the findings. Future studies should consider a larger and more diverse sample, including females and individuals with relevant clinical conditions.

Authors:  In the revised manuscript, we clarified that the recruitment of only males was due to the exploratory nature of the present study. We wanted to avoid the potential influence of hormonal fluctuations, such as those associated with the menstrual cycle in women, which could affect pain perception, as demonstrated in the literature (Yildirim et al., 2020; PMID: 31938578 for an example). In the Discussion, we emphasized that the present findings motivate future studies on both males and females. We also acknowledged that the limited and unpaired sample size may weaken the impact of the present findings. We now added a new section in the Discussion (“4.3 Study limitations”) where we discussed the study limitations.

  1. Control Conditions: While the experimental design includes control conditions, the description of these controls and their justification could be clearer. It is crucial to elaborate on how these controls adequately address potential confounding variables.

Authors: We added more details in paragraph “2.2 Experimental procedures” highlighted in yellow. They provide additional details and explanations on the selection and meaning of the control conditions.

  1. Equipment and Measurement Techniques: The description of EEG equipment and settings is adequate, but the justification for the chosen parameters, particularly the EEG frequency bands analyzed, could be strengthened.

Authors: In the revised Introduction, we expanded the wording on the role of EEG alpha oscillations in cortical sensory-motor processing and explained the rationale for using individual alpha frequency peak (IAFp) for banding EEG alpha frequency bands in the present study.

Results: The results are clearly presented, but the interpretation is cautious and somewhat limited:

  1. Statistical Power: Given the small sample size, the statistical power of the study is questionable.

Authors: In the revised manuscript, we acknowledged the use of a small and unpaired sample size and have expanded the discussion on its implications for interpreting the results in the “4.3 Study limitations” section. Additionally, the effect size (Wilcoxon’s r) in the last column of Table 2 provides a clearer understanding of the strength of our findings.

  1. Data Presentation: Some of the figures and tables are overly complex. Simplifying these or providing more detailed captions could help in better communicating the findings.

Authors: We revised iconography and legends as follows. Based on another Reviewer's suggestion, we added a new figure (new Figure 1) illustrating the experiments' flowchart. Figure 2 (now Figure 3) was changed to include a subtitle [i.e., Time (milliseconds)]; the white boxes were removed to simplify it. Now, intervals of 1 s are represented as explained in the caption, and the timing of each event (i.e., REST, ANT, EXE, auditory cue, and electrical stimulus) is indicated as well. Figure 4 was split into two figures (new Figures 5 and 6) to make it more readable. Notably, we added information to the legends to make them more comprehensive on key methodological aspects, experimental and control conditions, and core concepts conveyed.

Discussion: The discussion provides a good link between the findings and existing literature, but it can be improved in several ways:

  1. Broader Implications: Expand on the clinical implications, suggesting specific changes or adaptations in clinical practices or future research.

Authors: In the revised Discussion, a new section, “4.2 Clinical implications and future perspectives,” has been added to extend the significance that the present work may have for future and clinical practice. We agree that this study's clinical perspective may be of interest to clinical neurophysiologists and psychologists.

  1. Limitations: The discussion of limitations is somewhat superficial. Detailed limitations, including the EEG's spatial resolution and the study's validity need to be added in this section.

Authors: In the “4.3 Study Limitations” section, we remarked on and discussed the intrinsic weaknesses of the present exploratory study and how to take them into account in future validation studies.

Conclusions:The conclusions are appropriately cautious given the exploratory nature of the study. However, they could more strongly emphasize the need for further research in more clinically diverse populations

Authors: In the revised Conclusions, we suggested using the present methodological approach in patients with chronic pain conditions typically undergoing MVF-based neurorehabilitation, such as phantom limb pain (PLP) syndrome [Ramachandran & Ramachandran, 1997, PMID: 8637922; Finn et al., 2017, PMID: 28736545; Thogersen et al., 2020, PMID: 32574786 from the revised manuscript], post-stroke hemiparesis [Bae et al., 2012, https://doi.org/10.1589/jpts.24.1119; Bartur et al., 2018, PMID: 30194016; Dohle et al., 2009, PMID: 19074686 from the revised manuscript], and complex regional pain syndrome [McCabe et al., 2003, PMID: 12509620; Tichelaar et al., 2007, PMID: 17473633 from the revised manuscript] to understand better the effects of MVF-based neurorehabilitation in those relevant different pathological conditions.

Reviewer 2 Report

Comments and Suggestions for Authors

The manuscript provides an interesting approach for the analysis of pain modulating. But, in this paper, there are still some areas that need improvement.

1. It is recommended to provide a more details about the history of the development of MVF technology and its applications, to enhance the research's contextual framework and significance.

2. Result of the plagiarism report with a significant similarity rate of 35%. The paper is in need of revision.

3. The citation format for references does not conform to the MDPI standard.

4. The experimental procedure involves many steps. An overall flowchart  is advised.

5. If possible, provide a more in-depth explanation or theoretical model regarding how MVF influences pain perception and neural physiological mechanisms.

6. Discuss the limitations of the study, such as sample size, constraints in study design, or issues regarding the generalizability of results.

7. Why only male volunteers were recruited?

8. The order of the filter was not provided.

Author Response

The manuscript provides an interesting approach for the analysis of pain modulating. But, in this paper, there are still some areas that need improvement.

  1. It is recommended to provide a more details about the history of the development of MVF technology and its applications, to enhance the research's contextual framework and significance.

Authors: We thank Reviewer 1 for the constructive and punctual comments. The introduction has been thoroughly expanded as required, and more details on the MVF procedure have been added.

  1. Result of the plagiarism report with a significant similarity rate of 35%. The paper is in need of revision.

Authors: This point has been discussed with the Editorial Staff during the submission phase. The similarity rate originates from the preprint version of the article and from being part of the first author’s (MR) PhD thesis, which was published by Aalborg University. These documents can be found at the following links:

- Preprint article: https://urldefense.proofpoint.com/v2/url?u=https-3A__www.authorea.com_users_517916_articles_592275-2Dneurophysiological-2Doscillatory-2Dmechanisms-2Dunderlying-2Dthe-2Deffect-2Dof-2Dmirror-2Dvisual-2Dfeedback-2Dinduced-2Dillusion-2Dof-2Dhand-2Dmovements-2Don-2Dnociception-2Dand-2Dcortical-2Dactivation-3Fcommit-3D483f5cd420ed9d2126eebac8e4a04f790bb2f0f5&d=DwIFaQ&c=shNJtf5dKgNcPZ6Yh64b-ALLUrcfR-4CCQkZVKC8w3o&r=DC4uydPUFRRMd0i9UyBq4iO9S3TBJGwGmOc8TFMe_Dw&m=Au_kQMnEcRLvFv1bVI0KN5pEPe3VMpyNoiUWHjzkO9yPGRTfsPNLV1GY98NIgrzq&s=7uzlO-lukjP_izLVrNsoaG4digxyuMeUqa2DQgvclOg&e=

- PhD thesis: https://urldefense.proofpoint.com/v2/url?u=https-3A__vbn.aau.dk_ws_portalfiles_portal_549454198_PHD-5FMR.pdf&d=DwIFaQ&c=shNJtf5dKgNcPZ6Yh64b-ALLUrcfR-4CCQkZVKC8w3o&r=DC4uydPUFRRMd0i9UyBq4iO9S3TBJGwGmOc8TFMe_Dw&m=Au_kQMnEcRLvFv1bVI0KN5pEPe3VMpyNoiUWHjzkO9yPGRTfsPNLV1GY98NIgrzq&s=r2OPXghaE-e8S8v_rHDBANWTnzNp811QZtQdzq0tM9I&e=

  1. The citation format for references does not conform to the MDPI standard.

Authors: We apologize for not following the Authors’ instructions on the bibliographic reference format. The references have now been formatted according to the MDPI standards in the revised manuscript.

  1. The experimental procedure involves many steps. An overall flowchart  is advised.

Authors: A new Figure (Figure 1) shows the flowchart of the entire experimental session.

  1. If possible, provide a more in-depth explanation or theoretical model regarding how MVF influences pain perception and neural physiological mechanisms.

Authors: The Introduction has been overall rephrased and extended, including further explanation of the existing literature regarding MVF and how this has been studied during sensory-motor processing (including pain processing).

  1. Discuss the limitations of the study, such as sample size, constraints in study design, or issues regarding the generalizability of results.

Authors: In the revised Discussion, a new section, “4.3 Study limitations,” has been added to provide an extensive explanation of the study's limitations in relation to its exploratory nature and a few suggestions to strengthen this research line in future validation studies.

  1. Why only male volunteers were recruited?

Authors: Only male participants were selected in the present exploratory study to avoid the potential influence of hormonal fluctuations, such as those associated with the menstrual cycle in women, which may affect pain perception, as demonstrated in the literature (Yildirim et al., 2020; PMID: 31938578, for an example). We acknowledged this significant limitation in the revised Discussion and emphasized that future validation studies should enroll both male and female participants.

  1. The order of the filter was not provided.

Authors: In the revised Methods, the EEGLAB filtering details were reported in paragraph “2.4 EEG recording and preprocessing” as advised in Widmann, Schroger, and Maess, 2015 (https://doi.org/10.1016/j.jneumeth.2014.08.002). This information is highlighted in yellow as follows: “Following visual inspection, data were filtered with a zero-phase basic FIR filter (passband edges: 0.3-40 Hz, cutoff frequencies: -6 dB, filter order: 846, transition band: 1 Hz)”.

Reviewer 3 Report

Comments and Suggestions for Authors

“Neurophysiological oscillatory mechanisms underlying the effect of mirror visual feedback-induced illusion of hand movements on nociception and cortical activation".

 Strengths of the Article:

This manuscript investigates the impact of Mirror Visual Feedback (MVF)-induced movement illusion on pain perception and cortical limbic midline region activation, which are typically associated with pain processing, body representation, and interoception. The study hypothesizes that the MVF-induced illusion of hand movement may influence pain perception and the activation of midline cortical areas linked to pain processing, as demonstrated by alpha Event-related Desynchronization (ERD). This study employed electrical stimuli of a fixed intensity on the left hand of healthy adult males to generate painful and non-painful sensations during three conditions: unilateral right-hand movements with MVF illusion, and right and bilateral hand movements without MVF. Electroencephalographic (EEG) activity was recorded from 64 scalp electrodes during these events. The ERD of EEG alpha rhythms (8-12 Hz) was used to measure the neurophysiological oscillatory mechanisms that induce cortical activation.

The results indicated that, compared to painful sensations, non-painful sensations were associated with decreased alpha ERD in the cortical midline, angular gyrus, and lateral parietal regions during the MVF condition. Conversely, during the control conditions without MVF, non-painful sensations were linked to increased alpha ERD in the lateral prefrontal and parietal regions. These preliminary findings suggest that the MVF-induced illusion could influence pain perception and neurophysiological mechanisms by reducing activation in cortical limbic and default mode regions, which may affect self-perception, internal body awareness, and attention to electrical signals from the stimulated hand.

While the study provides valuable insights, these findings are preliminary and require further investigation to confirm their validity and potential application in developing beneficial strategies for patients with chronic pain and motor impairments. Furthermore, this manuscript needs significant improvements to enhance clarity and readability. The major critiques and suggestions are outlined below:

Major Critiques and Suggestions:

1. The study recruited only adult male volunteers. The rationale for this choice should be clarified, and future studies should consider including a diverse sample to improve generalizability.

2. The rationale behind the auditory cues (70 dB, 1000 Hz, and 50 ms) used is not clear. The manuscript should discuss the effect of variable frequency and durations and how these parameters were selected.

3. The description of the three different conditions (80 trials each) in randomized order is missing from the method section. This information needs to be included for better understanding.

4. Figure 4, depicting the spatial distribution of the voxel-by-voxel, should use a larger text font for improved readability.

5. The significance of this study and its novel contributions should be discussed, as many findings from this study are already known.

6. The limitations of the EEG techniques, such as their lack of fine spatial resolution, should be addressed in the discussion.

Minor Critiques and Suggestions:

The following references were not cited in the text. If they are not needed, remove them; otherwise, cite them appropriately:

[79] Heinrichs-Graham E, Kurz MJ, Gehringer JE, Wilson TW. The functional role of post-movement beta oscillations 667 in motor termination. Brain Struct Funct. 2017 Sep;222(7):3075-3086. doi: 10.1007/s00429-017-1387-1. 668

[80] Körmendi J, Ferentzi E, Weiss B, Nagy Z. Topography of Movement-Related Delta and Theta Brain Oscillations. 669 Brain Topogr. 2021 Sep;34(5):608-617. doi: 10.1007/s10548-021-00854-0. Epub 2021 Jun 15. 670

[81] Pellegrino G, Tomasevic L, Herz DM, Larsen KM, Siebner HR. Theta activity in the left dorsal premotor cortex 671 during action re-evaluation and motor reprogramming. Front Hum Neurosci. 2018;12:364. doi: 672 10.3389/fnhum.2018.00364. 673

[82] Saleh M, Reimer J, Penn R, Ojakangas CL, Hatsopoulos NG. Fast and slow oscillations in human primary motor 674 cortex predict oncoming behaviorally relevant cues. Neuron. 2010;65(4):461–471. doi: 10.1016/j.neuron.2010.02.001. 675

[83] Schramm S, Albers L, Ille S, Schröder A, Meyer B, Sollmann N, Krieg SM. Navigated transcranial magnetic 676 stimulation of the supplementary motor cortex disrupts fine motor skills in healthy adults. Sci Rep. 2019;9(1):17744. 677 doi: 10.1038/s41598-019-54302-y.

Comments on the Quality of English Language

Minor editing of the English language is required in the manuscript.

Author Response

Major Critiques and Suggestions:

  1. The study recruited only adult male volunteers. The rationale for this choice should be clarified, and future studies should consider including a diverse sample to improve generalizability.

Authors: In the revised manuscript, we clarified that the recruitment of only males was due to the exploratory nature of the present study. We wanted to avoid the potential influence of hormonal fluctuations, such as those associated with the menstrual cycle in women, which could affect pain perception, as demonstrated in the literature (Yildirim et al., 2020; PMID: 31938578 for an example). In the Discussion, we emphasized that the present findings motivate future studies on both males and females. We also acknowledged that the limited and unpaired sample size may weaken the impact of the present findings. We now added a new section in the Discussion (“4.3 Study limitations”) where we discussed the study limitations.

  1. The rationale behind the auditory cues (70 dB, 1000 Hz, and 50 ms) used is not clear. The manuscript should discuss the effect of variable frequency and durations and how these parameters were selected.

Authors: We enhanced the explanation of the selected auditory parameters at the beginning of section “2.2 Experimental procedure”. Specifically, we referenced a pioneering study in the field of auditory investigation using EEG technique that employed identical parameters (Williams et al., 2005; https://doi.org/10.1080/00207450590958475). Additionally, we cited a study (Justen & Herbert, 2016; PMID: 27777557) demonstrating that these parameters do not affect the activity of cortical areas adjacent to the auditory cortex, such as the motor areas, the inferior parietal lobule, and the anterior and posterior cingulate cortices, which were of particular interest in our study.

  1. The description of the three different conditions (80 trials each) in randomized order is missing from the method section. This information needs to be included for better understanding.

Authors: Authors: We added more details in paragraph “2.2 Experimental procedures” highlighted in yellow. They provide additional details and explanations on the selection and meaning of the control conditions. Furthermore, we added a flowchart of the entire experimental procedure that could help the understanding of the full procedure.

  1. Figure 4, depicting the spatial distribution of the voxel-by-voxel, should use a larger text font for improved readability.

Authors: To improve the readability of Figure 4, we split it into two separate figures. In the revised manuscript, Figure 5 represents the (eLORETA) EEG alpha source activity during the anticipation of the movement, whereas Figure 6 represents that activity during the execution of the movement.

  1. The significance of this study and its novel contributions should be discussed, as many findings from this study are already known.

Authors: In the revised Discussion, a new section, “4.2 Clinical implications and future perspectives,” has been added to extend the significance that the present work may have for future and clinical practice. We agree that this study's clinical perspective may be of interest to clinical neurophysiologists and psychologists.

  1. The limitations of the EEG techniques, such as their lack of fine spatial resolution, should be addressed in the discussion.

Authors: In the “4.3 Study Limitations” section, we remarked on and discussed the intrinsic weaknesses of the present exploratory study and how to take them into account in future validation studies.

Minor Critiques and Suggestions:

The following references were not cited in the text. If they are not needed, remove them; otherwise, cite them appropriately:

[79] Heinrichs-Graham E, Kurz MJ, Gehringer JE, Wilson TW. The functional role of post-movement beta oscillations 667 in motor termination. Brain Struct Funct. 2017 Sep;222(7):3075-3086. doi: 10.1007/s00429-017-1387-1. 668

[80] Körmendi J, Ferentzi E, Weiss B, Nagy Z. Topography of Movement-Related Delta and Theta Brain Oscillations. 669 Brain Topogr. 2021 Sep;34(5):608-617. doi: 10.1007/s10548-021-00854-0. Epub 2021 Jun 15. 670

[81] Pellegrino G, Tomasevic L, Herz DM, Larsen KM, Siebner HR. Theta activity in the left dorsal premotor cortex 671 during action re-evaluation and motor reprogramming. Front Hum Neurosci. 2018;12:364. doi: 672 10.3389/fnhum.2018.00364. 673

[82] Saleh M, Reimer J, Penn R, Ojakangas CL, Hatsopoulos NG. Fast and slow oscillations in human primary motor 674 cortex predict oncoming behaviorally relevant cues. Neuron. 2010;65(4):461–471. doi: 10.1016/j.neuron.2010.02.001. 675

[83] Schramm S, Albers L, Ille S, Schröder A, Meyer B, Sollmann N, Krieg SM. Navigated transcranial magnetic 676 stimulation of the supplementary motor cortex disrupts fine motor skills in healthy adults. Sci Rep. 2019;9(1):17744. 677 doi: 10.1038/s41598-019-54302-y.

Authors: The references mentioned above are reported in Supplementary Materials, and as required by the editorial rules “Citations and references in the Supplementary Materials are permitted provided that they also appear in the reference list of the main manuscript”.

Reviewer 4 Report

Comments and Suggestions for Authors

In this paper, the authors investigated the underlying mechanism of activation due to mirror visual feedback-induced illusion tasks. The paper is well-written and structured. The following are some minor comments to improve the manuscript further.

1)  Figure 2 does not clearly explain the paradigm. This figure can be further improved. What are the blank boxes showing? 

2) Figure 4 needs to be readable. More enhanced Figure should be added.

Author Response

In this paper, the authors investigated the underlying mechanism of activation due to mirror visual feedback-induced illusion tasks. The paper is well-written and structured. The following are some minor comments to improve the manuscript further.

1)  Figure 2 does not clearly explain the paradigm. This figure can be further improved. What are the blank boxes showing?

Authors: We thank Reviewer 1 for the constructive and punctual comments. We modified Figure 2 (now Figure 3, as a new Figure 1 has been added) by providing a subtitle [i.e., Time (milliseconds)] and by removing the white boxes that might result in misleading. Now, intervals of 1 s are represented as explained in the caption, and the timing of each event (i.e., Rest, ANT, EXE, auditory cue, and electrical stimulus) is indicated as well. We improved the legend to make the figure more readable.

2) Figure 4 needs to be readable. More enhanced Figure should be added.

Authors: To improve the readability of Figure 4, we split it into two separate figures. In the revised manuscript, Figure 5 represents the (eLORETA) EEG alpha source activity during the anticipation of the movement, whereas Figure 6 represents that activity during the execution of the movement.

Round 2

Reviewer 1 Report

Comments and Suggestions for Authors

Authors have addressed most of my comments. 

Comments on the Quality of English Language

Minor English language editing is recommended. 

Reviewer 2 Report

Comments and Suggestions for Authors

All revisions have been appropriately addressed and refined. I recommend proceeding with publication.

Reviewer 3 Report

Comments and Suggestions for Authors

I am delighted to note the authors diligent consideration of the reviewers suggestions, evident in the revised manuscript where they have effectively addressed all raised issues and incorporated the recommended changes. These revisions notably enhance the clarity and organization of the manuscript, particularly in navigating complex technical discussions, thereby improving accessibility and readability. This concerted effort contributes significantly to the overall quality of the article.

Comments on the Quality of English Language

Minor editing of the English language is required